# Towards Printed Pediatric Medicines in Hospital Pharmacies: Comparison of 2D and 3D-Printed Orodispersible Warfarin Films with Conventional Oral Powders in Unit Dose Sachets

**DOI:** 10.3390/pharmaceutics11070334

**Published:** 2019-07-14

**Authors:** Heidi Öblom, Erica Sjöholm, Maria Rautamo, Niklas Sandler

**Affiliations:** 1Pharmaceutical Sciences Laboratory, Åbo Akademi University, Artillerigatan 6A, 20520 Åbo, Finland; 2HUS Pharmacy, HUS Helsinki University Hospital, Stenbäcksgatan 9B, 00290 Helsingfors, Finland

**Keywords:** warfarin, 3D printing, semisolid extrusion 3D printing, inkjet printing, orodispersible film, oral powder, pediatric, hospital pharmacy, personalized medicine, on-demand manufacturing

## Abstract

To date, the lack of age-appropriate medicines for many indications results in dose manipulation of commercially available dosage forms, commonly resulting in inaccurate doses. Various printing technologies have recently been explored in the pharmaceutical field due to the flexible and precise nature of the techniques. The aim of this study was, therefore, to compare the currently used method to produce patient-tailored warfarin doses at HUS Pharmacy in Finland with two innovative printing techniques. Dosage forms of various strengths (0.1, 0.5, 1, and 2 mg) were prepared utilizing semisolid extrusion 3D printing, inkjet printing and the established compounding procedure for oral powders in unit dose sachets (OPSs). Orodispersible films (ODFs) drug-loaded with warfarin were prepared by means of printing using hydroxypropylcellulose as a film-forming agent. The OPSs consisted of commercially available warfarin tablets and lactose monohydrate as a filler. The ODFs resulted in thin and flexible films showing acceptable ODF properties. Moreover, the printed ODFs displayed improved drug content compared to the established OPSs. All dosage forms were found to be stable over the one-month stability study and suitable for administration through a naso-gastric tube, thus, enabling administration to all possible patient groups in a hospital ward. This work demonstrates the potential of utilizing printing technologies for the production of on-demand patient-specific doses and further discusses the advantages and limitations of each method.

## 1. Introduction

The lack of suitable dosage forms or doses for children is a common situation in hospital wards [1]. To date, there are no commercially manufactured age-appropriate oral formulations containing warfarin sodium (WS) available for children, especially for neonates and infants. Warfarin is an anticoagulant with a narrow therapeutic index [2,3], which for pediatrics is used to prevent and treat thrombotic events in identified risk groups such as patients suffering from cancer, short bowel syndrome, and patients with a central venous catheter for administration of total parenteral nutrition [3]. The therapy involves dose titration and monitoring of the International Normalized Ratio (INR) [2]. Achieving the right dose often requires manipulation of tablets as WS is commercially available in Finland only as conventional tablets in strengths of 3 and 5 mg. A common way of tailoring the dose in hospital wards is by splitting the tablet into halves or quarters, which then further may be crushed and dissolved or dispersed before administration if the patient is unable to swallow tablets or in cases where the drug is administered through a naso-gastric tube. Splitting of tablets into halves or even smaller parts does not always result in uniform pieces regarding weight and drug content, which might lead to variability in doses and absorbed drug amounts resulting in a risk for under- or overdosing [4,5,6,7]. An alternative to tablet splitting is compounding of oral liquids, capsules, or oral powders in unit dose sachets (OPS) at the hospital pharmacy [8]. The dosing flexibility is better for oral liquids than for OPS or capsules. A disadvantage of oral liquids is, on the other hand, the stability, which usually is shorter for liquid dosage forms than for compounded solid dosage forms. The compounding of oral liquids might also require the use of excipients that can be harmful to children. Oral powders, capsules, and segments of tablets are dissolved or dispersed in a liquid before administration or alternatively given with food. For many hospitalized children the medication is given through an enteral feeding tube, which require the drug to be in liquid form or formulated in such a way that it easily can be dissolved or dispersed prior to administration. The fact that some patients have fluid restrictions further affect the requirements for the administration of dosage forms.

A need for tailored doses as well as dosage forms that are easy to administer to children has led to the development of new formulations such as mini-tablets and orodispersible films (ODFs) [9,10,11]. Both dosage forms are shown to be suitable for infants aged 6–23 months and preschool children aged 2–6 years [12,13,14]. ODFs are thin, rapidly dissolving or disintegrating polymer films that are administered directly into the mouth where they stick to the tongue or palatal [9,15]. The administration does not require water intake, making it a good alternative for patients with fluid restrictions or patients that are unable to swallow conventional tablets [9]. Disintegration time, as well as, film thickness and stickiness, has been identified as key acceptability characteristics for ODFs in healthy young adults [15]. Different methods mentioned in the literature for the production of ODFs are solvent casting, hot-melt extrusion, semisolid casting, solid dispersion extrusion, rolling, and printing technology methods, such as flexographic printing, semisolid extrusion 3D printing (EXT), and inkjet printing (IJP) onto edible substrates [16,17,18,19,20].

Novel three dimensional (3D) printed dosage forms have been presented in the literature as a potential way of manufacturing accurate, personalized doses for pediatric patients in hospital pharmacy settings [21]. Individual preferences, for instance, size, color, and shape of the printed dosage form are features that affect the acceptability of a dosage form [22]. By printing personalized medicines and considering an individual child’s preferences as well as the need for individual doses, it would be possible to improve the patient centricity of hospital pharmacy compounding. 3D printing is a manufacturing technique that uses a computer-aided design (CAD) to deposit layers of material on top of each other producing 3D objects [19]. Examples of potential printing technologies for pharmaceutical manufacturing are flexographic printing, fused deposition modeling (FDM), EXT and 2D or 3D IJP [21,23,24]. Flexographic printing technology has successfully been used to produce drug-loaded ODFs [16]. FDM, also known as fused filament fabrication (FFF), is one of the most extensively investigated printing methods in the pharmaceutical field. The FDM process uses high temperatures to melt the thermoplastic drug-loaded polymer filament used as feedstock material, which is why the method may be unsuitable for thermolabile active pharmaceutical ingredients (API) and polymers [23,25]. Depending on the choice of polymer as well as the design of the dosage form, both immediate and sustained release formulations can be manufactured by means of FDM [26]. EXT, also called pressure-assisted microsyringe printing method (PAM), utilizes a semisolid formulation, e.g., gel or paste, as starting material [27,28]. The formulation is loaded in a syringe and extruded through the nozzle, by, e.g., pressurized air to form a solid dosage form. Immediate release tablets containing levetiracetam [28] and paracetamol [29] have been produced by means of EXT. The deposition of API containing ink onto edible substrates as well as jetting a binder solution onto a powder bed to form solid objects are two methods where IJP is explored for production of dosage forms [21,23]. Readers interested in more detailed information regarding printing methods explored in the pharmaceutical field are referred to the following reviews [21,30].

Recently studies have been conducted to produce personalized doses of warfarin by means of printing. Tian et al. have printed oral disintegrating tablets of WS using the binder jetting technique [31]. A binder liquid was sprayed from a nozzle on a powder bed, and the process was repeated in several layers until the desired dose of the tablet was achieved. Vuddanda et al. were able to produce ODFs with IJP technology [32]. A modified commercial printer was used to deposit a 300 mg/mL warfarin solution onto different sizes of edible substrates made of hydroxypropyl methylcellulose (HPMC), glycerol, and water, resulting in ODFs containing two different doses of warfarin. Additionally, EXT has been utilized to print ODFs containing various doses of WS using hydroxypropyl cellulose (HPC) as a film-forming agent [33]. Another approach to produce tailored doses of warfarin containing ODFs, by traditional solvent casting rather than utilizing printing techniques, has recently been presented by Niele et al. [34]. A long ODF placed in a tape dispenser enabled administration of personalized warfarin doses by tearing off a piece of ODF corresponding to a specific dose.

This study aims to compare the use of EXT and IJP with the conventional manufacturing method for compounding OPSs to produce various doses of warfarin. It further seeks to evaluate the quality and stability of the IJP and EXT printed ODFs as well as the OPSs prepared at a hospital pharmacy. To the best of our knowledge, there are no previous studies where the content uniformity of printed dosage forms would have been compared to traditional pharmacy compounded dosage forms like OPSs. The content uniformity and dose accuracy were expected to be better for the printed dosage forms than the oral powders as IJP is considered a very accurate method to prepare low-dose drugs, whereas OPSs or capsules have shown some inaccuracy in previous studies [20,21,35]. The ease of use was assessed by the suitability to administer the prepared dosage forms through a naso-gastric tube, describing how well the administration needs of the wards can be met. The risk for medication errors in drug administration was addressed by inkjet printing a QR code onto the ODFs. The final aim of this study was to evaluate the suitability of the utilized printing methods for extemporaneous compounding in a hospital pharmacy environment.

## 2. Materials and Methods

### 2.1. Materials

The anticoagulant warfarin was the drug investigated in the present study. Warfarin sodium (WS) loaded into the orodispersible films (ODFs) was acquired from Sigma-Aldrich (St. Louis, MO, USA), and the WS present in the compounded oral powders in unit dose sachets (OPSs) was obtained from commercial Marevan forte 5 mg tablets (Orion Pharma, Espoo, Finland). Lactose monohydrate (parve granules, Oriola, Espoo, Finland) was used as a filler in the OPS together with the ground tablets. Hydroxypropylcellulose (HPC, Klucel™ EXF, MW 80,000), which was used as a film-forming agent for both the EXT and IJP ODFs, was kindly donated by Ashland (Schaffhausen, Switzerland). Quinoline yellow (Sigma-Aldrich, Bangalore, India) and propylene glycol (PG) ≥ 99.5% (Sigma-Aldrich, St. Louis, MO, USA) were added to the IJP ink due to their respective properties as a colorant and viscosity/surface tension modifier. Ethanol ≥ 94% (Etax A, Altia, Helsinki, Finland) and purified water (Milli-Q water, Millipore SA-67120, Millipore, Molsheim, France) were used for analytics and as solvents in the polymer and ink solutions.

### 2.2. Methods

#### 2.2.1. Manufacturing of Personalized Doses

Target doses of 0.1, 0.5, 1, and 2 mg, were prepared by three different manufacturing methods. Two new innovative manufacturing methods in the pharmaceutical field, namely semi-solid extrusion 3D printing (EXT) and 2D inkjet printing (IJP), were compared to the established manufacturing method for oral powder unit dose sachets (OPSs) compounded at HUS Pharmacy, the hospital pharmacy of HUS Helsinki University Hospital (HUS) and subsequently used at New Children’s Hospital in Finland. The drug and the dose levels were selected based on an analysis of compounded OPSs at HUS Pharmacy in the year of 2018, which revealed that 1075 out of the 13,000 OPS manufactured at HUS Pharmacy contained the drug WS. The analysis additionally showed that the compounded WS doses were between 0.1 and 2.3 mg, where the most frequently compounded doses were 0.5 and 1 mg.

#### 2.2.2. Film Designs

Squared films with four different aimed doses were designed (Table 1) using a computer-aided design software (Inventor Professional software, version 2019, Autodesk, San Rafael, CA, USA) for the EXT ODFs and PowerPoint (version 2016, Microsoft Office, Microsoft Corporation, Redmond, WA, USA) for the IJP ODFs. EXT and IJP films were designed to have the same size, however, the printed area was designed to be slightly smaller for the IJP film to allow for manual cutting of the film after the printing step. A cutting template with the final size of the IJP film was also designed in PowerPoint. The EXT ODF designs were saved as .stl files and imported into the slicer software (RepertierHost v1.6.1, Hot-World GmbH and Co. KG, Willich, Germany) where the print settings were set, and the g-code was generated. The IJP designs were saved as .bmp files and imported into the printing software where the printing parameters were determined.

The film sizes were determined based on the assumption of what could be handled by a nurse at the hospital as well as physical considerations of pediatrics. The smallest film size was restricted by the size still manageable to handle, and the biggest film was limited by the size of a child’s mouth. The different sizes for the EXT ODFs were designed to increase in volume in the same ratio as the dose escalation in order to enable the use of the same printing solution for manufacturing of all sizes. The final sizes of the IJP ODFs were designed to be equal to the sizes designed for the EXT, as displayed in Table 1.

#### 2.2.3. Semisolid Extrusion 3D Printing

##### Printing Solution

The drug concentration in the HPC solution was determined by printing films (*n* = 6) of all designed sizes using a placebo HPC solution. The wet weight of the printed films was used to calculate the percent WS needed in the polymer solution to obtain the targeted doses. The average WS drug load for all of the different sizes based on the calculations was the selected drug load.

The placebo printing solution for the EXT was prepared by dissolving 15% (*w*/*w*) HPC in an ethanol and purified water mixture (ratio 1:1). The drug-loaded printing solution was prepared in a similar manner where 1.5% (*w*/*w*) WS and 15% (*w*/*w*) HPC were dissolved in a mixture of ethanol and purified water (ratio 1:1). The solutions were left on a magnetic stirrer overnight at room temperature to allow the polymer to fully dissolve.

##### Semisolid Extrusion 3D Printing

The prepared printing solutions were transferred into 10 mL disposable syringes attached to a single-use 25 G electro-polished tip (1/4″ Techcon TE Needle, Ellsworth adhesives, Norsborg, Sweden). The Biobots 1 (Biobot, Philadelphia, PA, USA) EXT equipped with an air compressor was used to print both placebo and drug-loaded ODFs. Films were printed on pieces of transparency sheets with a set pressure of 25 PSI and a printing layer height of 0.1 mm. One vertical shell was printed, and the outlines were subsequently filled in using a rectilinear fill pattern with a 45° fill angle, an infill density of 100% and an infill overlap of 15%. All printing steps were conducted with a speed of 8 mm/s. Within each batch, one film was printed at a time, and the films were let to dry overnight in room temperature.

EXT films were printed on three different days, referred to as batch 1, 2, and 3, to evaluate the day-to-day and batch variability of the manufacturing method. The same printing solution was used for printing of all batches in order to obtain information regarding the robustness of the technique rather than differences possibly originating from the preparation of the print solution. As the same solution was used for printing on three different days (day 1, 2, and 4), the stability of the drug-loaded printing solution stored in room temperature was determined by UV-spectrophotometry (Lambda 35, PerkinElmer, Singapore, Singapore) at 207 nm.

#### 2.2.4. Inkjet Printing

##### Preparation of Solvent Cast Printing Substrates

The polymeric substrates used in the IJP process were prepared by solvent casting. A drug-free HPC solution was prepared in the same manner as described for the placebo EXT ODFs and subsequently cast into films with a wet thickness of 600 µm utilizing a film applicator (Multicator 411, Erichsen, Hemer, Germany). The films were cast on top of transparency sheets (clear transparent X-10.0, Folex, Germany) and allowed to dry in room temperature minimum overnight (some longer, due to the printing of batches on different days).

##### Ink Formulation

The 100 mg/g WS ink solution for the IJP was obtained by dissolving WS (10% (*w*/*w*)) in the ink base consisting of a mixture of purified water (5% (*w*/*w*)), PG (27% (*w*/*w*)) and ethanol (57.99% (*w*/*w*)) (Table 2). The colorant quinoline yellow (0.01% (*w*/*w*)) was added to the ink solution in order to better visualize the printed area. A placebo ink was prepared in the same ratio as the drug-loaded ink, which was used for the production of placebo-imprinted ODFs needed for the analytics. Both inks were stored in the fridge at 4 °C. As for the EXT, the stability of the ink over multiple days was determined in a preliminary study by UV-spectrophotometry (Lambda 35, PerkinElmer, Singapore) at 207 nm.

##### Inkjet Printing

Inkjet printing was performed with a PixDro LP50 piezoelectric printer (Roth and Rau, Eindhoven, Netherlands) equipped with a print head with 128 nozzles (SL-128 AA, Fujifilm, Tokyo, Japan) and a camera for visualization of the jetted droplets. The printing resolution was set to 720 dpi based on calculations of the target dose, estimated droplet volume, ink concentration, as well as the size of the printed area. Printing was conducted with a jetting frequency of 1400 Hz, a voltage of 80 V, an ink pressure of –18 mbar and a pulse shape of 3-16-5 µs. The prepared ink (drug or drug-free) was filtered (0.45 µm polypropylene membrane syringe filter, VWR International, Radnor, PA, USA) and used to imprint the prefabricated solvent cast HPC films according to the premade designs using 40–60 nozzles, a quality factor of 3 and bi-directional printing. One printing run resulted in 32 printed films of a certain size that were allowed to dry in ambient conditions overnight and subsequently cut with a scalpel according to a template in order to obtain the final size.

#### 2.2.5. Compounding of Oral Powders in Unit Dose Sachets

The OPSs, each individual sachet weighing 200 mg, were compounded at the manufacturing unit at HUS Pharmacy in the same routinely manner as when OPSs are prepared and delivered for patients at the hospital. Three batches per dose were manufactured on three different days. The batch size was 30 OPSs except for the last batch of the 2 mg doses, where the batch size was 120 OPSs.

The OPSs were manufactured following the standard operating procedures of HUS Pharmacy for extemporaneously prepared OPSs. A pharmacist prepared the masses, and a technician or pharmacist weighed the individual sachets. Marevan forte 5 mg tablets were crushed in a mortar and ground with a pestle to a fine powder. Lactose monohydrate was added in geometric amounts to receive the final concentration and amount needed for each dose and batch size. The content of the individual sachets was weighed into waxed powder papers (Herra Järvisen Verstas Oy, Helsinki, Finland) using an analytical balance (MettlerToledo XP204, Greifensee, Switzerland). All sachets were labeled and packed in plastic ziplock bags.

#### 2.2.6. Identification Labeling using Printed Quick Response (QR) Codes

ODFs prepared by IJP and EXT were imprinted with a quick response (QR) code containing vital information about the dosage form, such as type of dosage form, API, strength, manufacturing date, expiration date, as well as the batch number. The QR code was generated utilizing the free online QR generator (goQR.me, Foundata GmbH, Karlsruhe, Germany), saved as a .bmp file and imported into the printing software. A placebo ink containing 1% (*w*/*w*) brilliant blue G dissolved in the ink base consisting of 27% (*w*/*w*) PG, 5% (*w*/*w*) purified water, and 67% (*w*/*w*) ethanol was used for printing the QR code.

The same IJP and print head as utilized for the printing of the IJP ODFs was used to print the QR code. The ink was filtered through a 0.45 µm polypropylene membrane syringe filter (VWR International, Radnor, PA, USA) to remove any undissolved particles and bi-directional printing was conducted with a pulse shape of 3-16-5, a jetting frequency of 1700 Hz, a voltage of 80V, and an ink pressure of −18 mbar. The QR code was printed with one nozzle, a quality factor 1 and a resolution of 400 dpi. The readability of the imprinted QR code on the ODFs was evaluated using QR reader for iPhone (version 6.8, Tapmedia Ltd., UK) and QR Code Reader (version 1.0.7, Google Commerce Ltd., Dublin, Ireland) for android.

#### 2.2.7. Weight, Thickness, and Appearance of Dosage Forms

The overall appearance of the prepared dosage forms was evaluated visually. The thickness of the ODFs was measured at 5 locations (all corners and the middle of the film) utilizing a caliper (CD-6”CX, Mitutoyo, Kawasaki, Japan) and the weight of the ODFs was determined using an analytical balance (AND GH-252, A and D Instruments Ltd., Tokyo, Japan). The weight and thickness of the ODFs used in the content analysis were chosen to represent the respective batches (average ± SD, *n* = 10).

#### 2.2.8. Mechanical Testing

The mechanical properties of the produced ODFs were investigated using a TA-XTplus (Stable Micro Systems, Godalming, UK) texture analyzer equipped with a 10 kg load cell. The largest ODFs (*n* = 5) were one at a time clamped between the Perspex film support platform and the aluminum circular top plate (Film support rig HDP/FSR, Stable Micro Systems). The spherical probe (ø 5 mm, SMS P/5S, Stable Micro Systems) was used to puncture the film with a constant speed of 1 mm/s until reaching the target distance of 5 mm (EXT films) or 15 mm (IJP films). The acquisition of data started when the trigger force of 0.049 N was reached, and the maximum applied force and penetration depth (mm) into the film before rupturing was recorded. Experiments were conducted at ambient conditions.

#### 2.2.9. Surface pH

The surface pH of the prepared EXT and IJP ODFs (drug-loaded and placebo) as well as of the prepared OPSs was determined in room temperature by placing one 2 mg dosage form in a small glass vial and adding 1 mL of purified water. The electrode of the pH meter (Mettler Toledo FE20, Mettler Toledo AG, Greifensee, Switzerland) was lowered into the solution, and the surface pH was determined after being immersed for 1 min and 15 min, respectively. Measurements for each formulation were performed in triplicate.

#### 2.2.10. Moisture Content

The moisture content of the prepared dosage forms (*n* = 3) was investigated utilizing a moisture analyzer (Radwag Mac 50/NH, Radom, Poland). The sample with a target dose of 2 mg was placed on an aluminum pan, and the mass % weight loss corresponding to moisture evaporation was recorded as the sample was heated up to 120 °C. The end point of the measurement was set to when the change of mass had reached equilibrium and was less than 1 mg/min.

#### 2.2.11. Disintegration

The disintegration time of the ODFs was investigated using the Petri dish method. The films were analyzed with regards to thickness and weight prior to the disintegration test. 10 mL of purified water was pipetted into a Petri dish, and the ODF was subsequently dropped on top of the liquid surface using tweezers. The time for the film to completely rupture in the middle into smaller film pieces utilizing this static method was recorded and reported as the time for the film to disintegrate. In other words, swelling (in any direction) of the film or small pieces wearing off at the edges was not defined as the endpoint.

#### 2.2.12. Drug Content

Drug content of the prepared doses was determined to evaluate the amount of drug obtained in the final dosage form utilizing the different manufacturing techniques. Briefly, one ODF or OPS was placed in 100 mL of purified water and shaken (Multi-shaker PSU 20, Biosan, Latvia) at 50 rpm for a minimum of 3 h. Samples were diluted when necessary, and the absorbance was subsequently spectrophotometrically (Lambda 35, PerkinElmer, Singapore, Singapore) analyzed at 207 nm. The absorption of the drug-free ODFs at 207 nm, consisting of either a HPC film or a HPC placebo-ink imprinted film, was used as a baseline for the measurements. For the filtered (0.2 µm cellulose acetate syringe filter) OPSs, the absorbance of purified water at 207 nm was considered as the baseline. Ten replicates of all prepared doses were analyzed for each batch and manufacturing method. For stability, ten replicates of the largest target dose were analyzed at each stability time point.

Uniformity of content of single-dose preparations (UC) was calculated according to the European Pharmacopeia (Ph. Eur. 9.0) 2.9.6, test B [36]. The test complies with requirements if not more than one individual content is outside 85 and 115% of average content, and none is outside 75 and 125% of average content. If two or three individual dosage units are outside 85–115% of average content, a further 20 units should be tested. The test fails to comply with requirements if more than three individual contents are outside 85–115% of average content. Moreover, the prepared dosage forms were analyzed with regards to uniformity of dosage units as described in Ph. Eur. 2.9.40. The acceptability constant k = 2.4 (*n* = 10) and T = 100% were used to calculate the acceptance values (AV). The AV (L1) should be ≤ 15.0 to meet the requirements. In this study, the acceptance UC and AV were based on ten replicates, an additional 20 dosage forms were not analyzed.

#### 2.2.13. In Vitro Dissolution

In vitro drug release studies were conducted for the pure drug as received, as well as for the prepared doses from the three different manufacturing techniques (EXT, IJP, and OPS) to study the drug release behavior of the dosage forms. The thickness and weight of the ODFs was documented prior to the dissolution study as well as the weight of the OPSs. The ODFs were placed in dissolution baskets and inserted in 250 mL glass bottles containing 100 mL of purified water, while the oral powder in the sachets were emptied from the sachets directly into the bottles. The bottles were kept on a shaking water bath at 37 °C and 50 rpm throughout the dissolution study. At each predetermined time point, 3 mL of media was manually withdrawn, and 3 mL of fresh media was added. The absorbance of the withdrawn solutions was measured at 207 nm using a UV–VIS spectrophotometer (Lambda 35, PerkinElmer, Singapore). The withdrawn solutions from the OPS samples were filtered through a 0.2 µm cellulose acetate syringe filter (rinsed with 30 mL of purified water prior to use) in order to remove undissolved particles. Samples were measured in triplicate, and the percent drug released was calculated based on the results obtained from the content measurements.

As a comparison to the manual dissolution, dosage forms with the highest drug load (target dose of 2 mg) were additionally studied utilizing an automated setup (Sotax AT 7smart, Basel, Switzerland). The ODFs were accurately weighed an inserted into baskets, while the oral powders from the OPSs were directly poured into the vessels filled with 500 mL of purified water at 37 ± 0.5 °C. The basket rotated with a speed of 50 rpm, and samples of the release media were automatically withdrawn at predefined time-points with the use of a pump (Sotax CY 6, Basel, Switzerland), filtered (glass microfiber filter GF/B, GE Healthcare Life Sciences, Little Chalfont, UK) and the absorbance measured at 207 nm utilizing an on-line UV–VIS spectrophotometer (Lambda 35, PerkinElmer, Singapore). The average percent drug release (*n* = 3) was once again calculated based on the results from the content measurements.

#### 2.2.14. Evaluation of Drug Administration through a Naso-Gastric Tube

The administration of the produced dosage forms through a naso-gastric tube was mimicked to ensure that it would be possible to administer the prepared dosage forms to all patients at hospital wards. The amount of water used for administrating one OPS dose is not clearly standardized at HUS, but typically, the volume is as small as possible. To simulate the process used at the hospital ward, each dosage form was placed in a disposable plastic medicine cup and 2 mL purified water was added. The medicine cup was shaken for approximately 2 min whereafter the solution was administered into the naso-gastric tube (Nutricia Flocare pur tube, CH 6/60, inner diameter 1.1 mm, Nutricia Medical Devices BV, Zoetermeer, Netherlands) with the help of a disposable syringe and subsequently collected into a 100 mL volumetric flask. After administration of the dosage form, the naso-gastric tube was flushed with 2 mL of purified water, which likewise was collected in the volumetric flask. Purified water ad 100 mL was added, and the WS content was measured utilizing the same UV–VIS spectroscopy method as described in drug content measurement (Section 2.2.12). Three replicates of the largest target dose for all three manufacturing methods were tested.

#### 2.2.15. Attenuated Total Reflectance Fourier Transform Infrared (ATR-FTIR)

The infrared spectra of the raw materials, physical mixtures, and the prepared dosage forms were obtained using an Attenuated Total Reflectance Fourier Transform Infrared (ATR-FTIR) spectroscopy (Spectrum Two, PerkinElmer Inc., Beaconsfield, UK). The samples were placed on top of the diamond (IJP ODFs with the printed side facing the diamond), and a force of 75 N was applied during the measurement to attain a good signal. Samples were measured from 4000 to 400 cm^–1^ with four accumulations at a resolution of 4 cm^−1^. Spectra were obtained in duplicate, and a third measurement was performed in cases where differences were observed during the first two measurements. The software Spectrum (version 10.03.02, PerkinElmer) was used for acquisition of the spectra and for further data treatment utilizing baseline correction, normalization, and data tune-up.

#### 2.2.16. Differential scanning calorimetry (DSC)

Differential scanning calorimetry (DSC) was utilized to evaluate the thermal properties of the samples using the Q2000 (TA Instruments, New Castle, DE, USA). Samples weighing 3.0 ± 0.1 mg were analyzed in sealed Tzero aluminum pans from –20 to 230 °C with a heating rate of 10 °C/min. The OPSs were only heated up to 220 °C to avoid further degradation of the sample. Measurements were performed in duplicate and in triplicate if differences were observed during the first two runs. Nitrogen was used as purge gas with a flow rate of 50 mL/min during all measurements. The data was analyzed utilizing the TA Universal Analysis software (version 4.5A, TA Instruments).

#### 2.2.17. Stability

The stability of the EXT ODFs, IJP ODFs, and OPSs was investigated by visual inspection, mechanical analysis (of ODFs), UV–VIS spectroscopy (drug content), DSC, and ATR-FTIR. The EXT ODFs were stored in a Petri dish throughout the stability period and the IJP ODFs sheets were stacked on top of each other with a transparency sheet in between the printed samples and further covered with aluminum foil. The OPSs were stored in open ziplock bags as an external package. All samples were stored in ambient conditions in a cupboard protected from light. The temperature and relative humidity was tracked during the period using a humidity and temperature USB data logger (wk057, Wisemann Klein SL, Barcelona, Spain). Samples were analyzed at predefined time points, namely at day 1, 7, 14, 21, and 28.

## 3. Results and Discussion

### 3.1. Manufacturing of Personalized Doses

In the initial phase of the study, 60 formulations (data not shown) were screened with regards to their film-forming capacity as well as suitability to be processed into personalized ODFs by means of printing. A film-forming solution that was suitable for both the utilized printing techniques was desired in order to identify differences in the final dosage form originating from the printing methods rather than the formulation itself. A simple formulation consisting of the drug and HPC dissolved in a mixture of purified water and ethanol was chosen due to the excellent film-forming capacity of HPC, which resulted in clear, flexible films without the need of plasticizers. The printed formulations consisted of as few excipients as possible as a preference expressed by the medical doctors at HUS, however, disintegrants, saliva stimulating agents, sweeteners, taste masking agents, flavors, colorants, etc., may be introduced in the formulation to further tailor the properties of the ODF or to fulfill individual preferences of a patient. The amounts used of these types of ingredients are typically quite low in ODFs [37,38,39], and it is, therefore, unlikely that addition of these materials dramatically would change the printability nor the quality of the printed ODFs. As EXT and IJP nonetheless are techniques that require different properties of the formulation to be printable, the idea was to use the same HPC solution (15% *w*/*w*) as a substrate for IJP which subsequently was imprinted with a drug ink and which in the case of EXT was drug-loaded and printed into ODFs in a single step.

The OPSs were produced according to the standard operating procedure at HUS Pharmacy in order to be able to compare the OPSs in use at the hospital with the recently explored innovative 2D and 3D printing techniques.

#### 3.1.1. Semisolid Extrusion Printing

The different sized ODFs were successfully manufactured according to the pre-made design utilizing EXT (a video of the printing process can be found in Appendix A). The drug-concentration required in the printing solution was calculated based on the wet weight of the printed placebo ODFs and the target dose of the ODFs. Six films of each size (target dose of 0.1, 0.5, 1, and 2 mg) were printed and immediately weighed (Table 3). The average calculated drug concentration of the four different sizes was selected as the drug concentration in the solution used for EXT printing. The correlation between the wet weight and size (mm^3^) of the printed placebo films as well as the subsequently dried films was 1 and 0.9998, respectively, suggesting that the weight of the prepared ODFs could be a reliable and easily accessible quality assurance method that could be utilized in a hospital setting as further confirmed in the drug content section.

The used EXT utilizes pressurized air to force the solution out of the syringe. In this study, the aimed pressure was 25 PSI and the actual pressure was noted to be 24.8 ± 0.1 at the beginning of the printing process (when pressing start). One discovered drawback with the used printer was that it was difficult to attain the set pressure and even during printing of a single ODF the pressure would typically fluctuate. As pressure is one of the most important parameters to determine how much material is deposited per unit time, it may result in ODFs with fluctuating drug amount. Other factors to take into account when using an EXT 3D printer is that the distance between the syringe tip and the build platform will have an impact on the amount of solution that is being deposited. Furthermore, the length of the tip and the amount of solution in the syringe was seen to have an effect on the pressure required and the amount of solution being deposited. Consequently, at least all of these factors should be standardized or monitored to achieve ODFs with similar properties.

#### 3.1.2. Inkjet Printing

The IJP ODFs were successfully produced in three subsequent steps involving, solvent casting of the substrate, deposition of the drug-loaded ink on the dry cast polymer sheet by means of IJP and finally drying overnight and cutting into the final size using a cutting template and a scalpel. A slightly modified ink from Genina et al. [40] was developed according to the requirements of the added components as well as the printer. A high concentration WS ink (100 mg/g) was used to enable printing of the desired dose in a single layer rather than using a multiple printing cycle approach as printing of a single layer was expected to decrease the manufacturing time and simultaneously act as visual quality control to spot non-printed areas. To achieve the target dose by printing a single layer, the dpi was calculated as described in the methods section. No clogging of the nozzles was observed during printing with the described ink formulation, even though recrystallization during printing of high concentration inks containing solvents that are easily evaporated may be of concern for IJP [40].

#### 3.1.3. Manufacturing Times

One of the cornerstones of personalized medicine is that a single or a few doses could be tailored according to a patient’s need at a specific time. Batches in personalized medicine are, therefore, typically small, as dose adjustments frequently may occur in order to achieve better treatment outcomes [26]. The manufacturing times for the different manufacturing methods were, hence, recorded to get an understanding of how time-consuming the production of a single dose would be.

The comparison of manufacturing times is somewhat challenging as the different techniques require different steps. In the case that these novel manufacturing techniques would become established manufacturing methods in, e.g., a hospital pharmacy setting, it would be desirable that the substrates, printing solutions, etc., would be contract manufactured and delivered to the hospital, where minor preparation steps such as addition of the desired API could be done. Based on that assumption, the noted manufacturing times in this study were the actual time it took for the printer to print the drug-loaded ODFs (not the preparation of, e.g., printing substrate or solution) whereas for the OPSs the manufacturing time included all the steps included in the SOP. The time to print a single ODF utilizing the two different printing techniques are shown in Table 4. The EXT ODFs were printed one at a time and the time was recorded from pressing start until the print head was returned to its starting position. The manufacturing time for a single IJP ODF was calculated based on all the printed batches consisting of 32 doses per batch.

The time required to print the IJP ODFs depended on the amount of used nozzles, in most cases so that an increased amount of used nozzles would lead to a decreased printing time. However, especially when printing the smaller sizes also the specific nozzles selected were observed to have an impact on the printing time as the print head in certain cases needs to move further to be able to use the chosen nozzles, thus understandably increasing the print time. Recording of manufacturing times showed that IJP was a faster technique than EXT, however, factors such as maintenance and cleaning require more time for the IJP compared to the EXT at least in a laboratory setting when the printer is not continuously running. The EXT printer utilizes disposable syringes and does not require a cleaning procedure in the same extent as the IJP printer, which needs to be flushed with a suitable solvent after use in order to ensure that contamination between different drugs and/or formulations do not occur. Printing of multiple films at once with the used EXT printer will reduce the manufacturing time as the transparency sheet does not need to be changed in between. It was furthermore noticed that the printing time for the smallest EXT ODFs with a target dose of 0.1 mg was, in total, 42 s from pressing start until the print head was returned to its starting position, where the actual print time stood for only 10 s. Printing of multiple EXT ODFs simultaneously may, however, be more challenging than printing only one ODF at a time due to, e.g., an unleveled build platform. This may affect the print quality and could result in increased fluctuations between the manufactured ODFs. Additionally, the print area in the used printer is relatively small, restricting the amount of ODFs printed at once.

One batch of OPSs consisted of 30 unit dose sachets except for the last batch of 2 mg OPSs, where the batch size was 120 units. Measured manufacturing times included the time to prepare the powder mass as well as the time to weigh all individual doses into powder papers and subsequently closing and labeling them. Preparation of the total mass for one batch of OPSs took 11.4 ± 2.6 min. The batch size did not affect the time needed to prepare the mass. Weighing of the OPSs took 31.7 ± 7.2 min for a batch of 30 units, whereas the time increased to 100 min for the batch of 120 units. Preparing dose powders in unit dose sachets does not require any premanufacturing steps or laborsome cleaning procedures as the manufacturing equipment, such as the pestle and mortar, are machine-washed after use.

#### 3.1.4. Identification Labeling Using Printed QR Codes

Dried EXT and IJP ODFs were successfully imprinted with readable QR codes (video of the printing process can be found in Appendix A) revealing the suitability to utilize IJP for enclosing information regarding the dosage form as also previously shown by Edinger et al. [41]. The QR code was printed directly on the ODFs with edible ink (Figure 1). QR codes are already in use for dosage forms prepared at HUS Pharmacy, but until now containing a limited amount of information (product number and batch number) and only present on the batch-specific label on the secondary package, not on a single dosage form.

The information incorporated in the QR code could easily be tailored according to the requirements or desires of the hospital. When manufacturing on-demand dosage forms tailored according to the need of the patient, patient information could beneficially be included in the QR code. An example of information that could be included is demonstrated in Figure 2. Incorporation of patient information in the QR code of the dosage form would allow the nurse to scan the code as an additional patient safety measure prior to administration to the patient. By doing that, it could, e.g., be ensured that the dose is intended for the specific patient, that the medication still should be given to the patient and the nurse would easily access information regarding how the doctor intended the dosage form to be administered. As patient information is classified as sensitive information, it should be handled securely according to the latest national legislation and general data protection regulations (GDPR). QR codes containing any sensitive information, such as patient information, should, therefore, have restricted access. In theory, this could be solved by the use of passwords and linkage to the patient database, which would require the same login information as otherwise used to access patient information. By linking the QR code, for example, to a database, it opens up the opportunity to include an increased amount of information without the QR code itself getting too detailed and, thus, minimizing the risk for the code to not be easily readable. Readers interested in the benefits and opportunities of QR-encoded dosage forms outside the hospital setting are referred to the interesting discussion by Edinger et al. [41].

### 3.2. Physical Appearance and Mechanical Properties of the Dosage Forms

The ODFs prepared by means of EXT resulted in clear, thin films with a slight wavy structure if inspected closely. The small waves originated from the printing process where the ODFs were built up one line at a time subsequently adhering to the previous and resulting in the final ODF. The weight and thickness of the respective batches and sizes can be found in Table A1. The IJP ODFs were slightly thicker than the EXT ODFs due to the nature of IJP. The printing substrates were cast with a wet thickness of 600 µm, resulting in films that were slightly thicker than the dried EXT ODFs in order to be able to absorb the jetted ink without dissolving the substrate. As a result, the IJP ODFs were also found to weigh more. The manual cutting of the IJP films into their final size resulted in surprisingly small weight differences of the films within a batch. However, a difference in weight should not have an impact on the drug content in the IJP ODFs as they were designed to have a drug-free area around the film, as demonstrated in Table 1. The OPSs consisted of a powder mixture of pink larger particles (ground tablets) and white smaller lactose monohydrate particles. Pictures of all the prepared dosage forms can be seen in Figure 3.

The mechanical properties of the prepared ODFs were investigated as ODFs should possess sufficient handling properties to ensure that they are not damaged during any of the steps involved preceding the administration of the dosage form [42]. Figure 4 displays the measured burst strength representing the maximum tolerated force (N) on the ODF before rupturing as well as the burst distance (mm) representing the flexibility of the ODF. The strongest ODFs were found to be the placebo IJP ODFs, which were prepared by solvent casting. The placebo IJP ODFs were observed to withstand over two times more force before rupturing compared to the placebo EXT ODFs, which may be explained by the fact that the placebo IJP ODFs also revealed a greater thickness. Moreover, the manufacturing method of the two placebo films differs, allowing the IJP ODF to be perfectly flat while the EXT ODF is built up one line at a time making the film slightly wavy. This may result in a more brittle film but, on the other hand, enable other advantages, such as fast disintegration compared to the solvent cast film (placebo IJP ODF) as will be discussed later.

Incorporation of the drug in the EXT ODFs appeared to only slightly decrease the strength of the ODFs, however, the burst distance decreased, and they were found to be the most brittle ODFs in this study. The ODFs to elongate the greatest before rupturing were the drug-loaded IJP ODFs. One-day-old WS IJP ODFs revealed an elongation distance of 5.61 ± 0.35 mm before rupturing, which is more than double (2.6 ± 0.2 mm) what was measured for the placebo IJP ODFs that served as the printing substrate. Introduction of the printing ink, hence, clearly showed an impact on the mechanical properties of the IJP ODFs. Additional moisture as well as PG, which both are known to have a plasticizing effect, seems to explain this behavior. Plasticizers typically interact with the polymer chains present in the formulation resulting in increased chain mobility and, consequently, a decreased glass transition temperature, which in terms of mechanical properties is seen as ODFs with improved plastic and elastic properties [43]. The drug-loaded EXT ODFs were much more brittle, bending 1.2 ± 0.1 mm (at day 1) before breaking, further indicating that both the manufacturing technique as well as the additional liquid applied during IJP influences the mechanical properties of the ODFs. The thickness of the films will to some extent have an impact on these results, but as can be seen in Table A2, the film thickness alone is not the reason to the differences between the formulations.

No major differences in the mechanical properties of the ODFs were witnessed during the one-month follow-up period for the EXT ODFs. Nevertheless, a tendency where the drug-loaded EXT ODFs show a decreased burst strength and burst distance one week after manufacturing may indicate that the ODFs were not completely dry after allowing to dry one day in room temperature. Another suggestion is that the films mechanical properties change according to the relative humidity. This can, however, not be supported by the placebo EXT ODFs and further studies are required to fully understand this phenomenon. The drug-loaded IJP ODFs show a trend where they over time become somewhat more brittle and simultaneously can resist a greater force before breaking, possibly due to slow evaporation of the deposited ink at room temperature. The conclusion of the mechanical study is that all prepared ODFs were possible to handle without breaking the dosage forms, however, further modifications such as changing the ODF thickness, addition/removal of plasticizer, etc., can be made to alter the properties of the ODFs if desired.

### 3.3. Surface pH

The surface pH of ODFs is commonly investigated to ensure that the pH of the dosage form is in the range of physiological saliva (5.8–7.4). An ODF with a pH largely differing from this may cause local mucosal irritation and discomfort for the patient [39]. The ODFs prepared in this study all revealed a neutral pH (Table 5) indicating that no local side effects should be encountered at the administration site. The OPSs showed a pH of 9.36 ± 1.62 after being in contact with water for 1 min, perchance making it too alkaline for pleasant administration of the dose. However, the pH decreased and was in the neutral range (7.34 ± 0.19) after allowing the OPS to further dissolve for 14 min suggesting that the OPSs should be fully dissolved/dispersed before administration to avoid possible irritation when administered with a small amount of water.

### 3.4. Moisture Content

The moisture content of pharmaceutical products is important to investigate as moisture present in dosage forms may have a negative effect on the physico-chemical, chemical, as well as microbiological stability of the final product [44]. However, some moisture present in ODFs is typically desired due to the plasticizing effect of water, as completely dry films tend to be brittle and have reduced handleability. Depending on the film-forming polymer used, excipients such as glycerol, PG, polyethylene glycol, sorbitol, macrogols of low molecular mass, citrates, and phtalates may be added due to their plasticizing effect of the ODFs [45,46]. However, it is important to keep in mind that plasticizers may alter the properties of the ODFs in multiple ways, e.g., the taste and mechanical properties of the ODF may be affected. Additionally, plasticizers have a tendency to absorb water, which is why a very high amount of plasticizer added to the formulation may result in similar stability issues as discussed above due to the absorbed water. ODFs with a too high a moisture content have also been described as sticky [39].

In this study, the moisture content of the dosage forms was studied using a moisture analyzer. The moisture content in the studied samples may originate from residual solvent from the manufacturing process or due to the hygroscopic nature of one or multiple components present in the formulation. The prepared dosage forms showed a mass loss of 9.36 ± 2.53% (EXT drug-loaded ODFs), 10.5 ± 2.45% (EXT placebo ODFs), 12.39 ± 1.68% (IJP drug-loaded ODFs), 9.13 ± 4.05% (IJP placebo ODFs) and 2.38 ± 0.26% (OPSs), as shown in Figure 5. As expected, the ODFs possessed a higher moisture content than the OPSs due to the introduction of moisture during the manufacturing process. Moreover, the ODFs were dried and stored in ambient condition allowing the hygroscopic HPC to absorb moisture from the surrounding air [47]. The drug-loaded IJP ODFs revealed the highest moisture content, which can be described by the fact that these ODFs consisted of a substrate corresponding to the placebo EXT ODF, which subsequently was imprinted with the drug-loaded ink. In addition to introducing further liquid to the dosage form, the ink contained 27% (*w*/*w*) PG, which is known for its plasticizing effect and therefore also its ability to hold water and it has been reported that ODFs with plasticizers, such as PG, showed increased moisture uptake [48]. Consequently, the drug-loaded IJP ODFs would likely require a longer drying time to allow further evaporation of solvents. The difference in moisture content between the placebo and drug-loaded IJP ODFs may, furthermore, be explained by the fact that the IJP placebo ODFs were prepared in advance and had an increased drying time compared to the drug-loaded IJP ODF that only dried overnight. Further studies would be needed to get in-depth information regarding the most suitable drying time and conditions for the prepared ODFs regardless of the manufacturing method.

Nair et al. [49] have stated that an ideal ODF should have a moisture content of less than 5%. In this study, all of the prepared ODFs failed to comply with that limit. However, the EXT ODFs and IJP ODFs were easy to handle and did not stick to gloves during handling. The IJP ODFs were observed to slightly stick to the packing material (transparency sheet between the printed sheets) indicating that the drying time preferably should be increased conceivably due to the high amount of PG present in the ink as discussed above. However, the brief stability study conducted did not indicate degradation of the drug, suggesting that the amount of moisture present in the formulations did not affect the stability of the drug during the studied period.

Lactose monohydrate is a substance with low hygroscopicity meaning it has a low tendency to absorb moisture from the surroundings. It typically has a moisture content of about 5%, which is in agreement with the low moisture content of the OPSs (2.38 ± 0.26%), as they largely consist of this filler. The water present in lactose monohydrate being tightly bound in the crystal lattice makes it chemically inert and not likely to interact with the drug present [50], indicating that moisture should not be a problem for the compounded OPSs.

### 3.5. Disintegration of Orodispersible Films

Rapid disintegration of ODFs is crucial due to the nature of the dosage form. Thus far, there are no specified methods or acceptance values available in the Ph. Eur. for testing the disintegration behavior of ODFs [36]. The limits available for orodispersible tablets (ODTs), stating that ODTs should disintegrate within 180 s, have therefore generally also been used for the novel ODFs [51]. In this study, all prepared ODFs complied with the disintegration limit described (Table A3). The different sizes of the EXT ODFs all disintegrated in less than 40 s. The IJP ODFs revealed an increased disintegration time (> 84 s), which can be explained by these ODFs being thicker compared to the EXT ODFs. Another difference originates from the manufacturing method, where the EXT ODFs have small waves originating from how the rectilinear infill was printed. These small lines likely result in an increased surface area and faster wetting of the ODF as compared the completely flat solvent cast film. The same phenomena, where a fast disintegration of fused deposition 3D-printed ODFs compared to solvent cast ODFs has recently been reported [52]. The drug-loaded ODFs required slightly longer time to disintegrate than the drug-free ODFs for both manufacturing methods, which may be explained by the presence of the drug in the formulation as also previously reported [37]. No major difference in the disintegration behavior of the ODFs was observed during the stability study (Table A4).

The OPSs could not be analyzed with this disintegration method as the dosage form already is in powder form. An attempt was nonetheless made to use the same method to identify the disintegration time of the ground Marevan particles as their pink color easily could be identified in the powder blend. The particles did not fully dissolve within 10 min and, therefore, the experiment was stopped. However, the Marevan particles dissolved instantly when the Petri dish was shaken after the experiment was stopped.

It is worth noticing that the disintegration method used in this study is a static method and the ODFs would likely disintegrate faster upon shaking. Additionally, the utilized Petri dish method allows the ODF to be wetted from only one side, which possibly results in slower disintegration. However, the method was selected, as no special equipment was needed. Furthermore, the film did not stick to the surface of the Petri dish with the chosen method, which might otherwise give false results. Since there is no standardized method in the Ph. Eur. it makes it difficult to compare disintegration results from different studies. Most importantly, even though a static method was selected in this study, all ODFs disintegrated within 180 s.

### 3.6. Drug Content

The drug content was determined for 10 dosage forms of each dose, batch, and manufacturing method and the measured content was compared to the target dose (Figure 6). Uniformity of content of single-dose preparations (UC) was determined in order to evaluate the precision of the used methods. Moreover, the acceptance value (AV) according to the Ph. Eur. was calculated, where dosage forms showing AV values of ≤ 15 comply with the set limit. Independent of the manufacturing method, all dosage forms prepared in the various batches were shown to comply with the limits stated for UC in the Ph. Eur. for the two largest doses containing 1 and 2 mg of WS (Table A1). Additionally, all ODFs passed the test for UC for the 0.5 mg dose, whereas one batch of the OPSs failed the test, having one unit outside ±25%, and another batch would have required testing of additional 20 OPSs since two individual contents were outside ±15% limits. For the 0.1 mg dose, two out of three batches for both EXT and IJP ODFs, as well as OPSs, fulfilled the requirements for the UC. For both the EXT and IJP ODFs the largest deviation from average content was more than 25% in one batch resulting in failure to comply with the set limit. For one batch of OPSs, an additional 20 units should have been analyzed to determine if the particular batch would pass or not. These results reveal that dose fluctuation occurs for low-dose dosage forms, but as the dose increases, all the studied manufacturing methods were able to produce repeatable dosage forms within the batch. However, between batches, differences in the average drug content and drug amount compared to the target dose were seen for all the manufacturing methods, as shown in Table A1. For the smallest dose (0.1 mg) the average drug content was 0.10–0.14 mg for EXT ODFs, 0.07–0.09 mg for IJP ODFs, and 0.05–0.09 mg for OPSs. Expressed as percentage of drug amount compared to target dose, the amount of WS varied between 72% and 140% for the EXT ODFs, between 52% and 112% for the IJP ODFs and between 36% and 104% for OPSs for the 0.1 mg dose when taking into account all the individual dosage forms manufactured in all three batches. This shows that in case of low-dose WS dosage forms, there can be a large fluctuation in the received dose for the child between the same dosage form prepared in different batches, which greatly may impact the treatment outcome. As the dose increases, the accuracy compared to the target dose also improves, and the fluctuation diminishes. For the 2 mg WS dose, the amount of drug compared to target dose was between 100.5–109.3% for EXT ODFs, 93.1–109.4% for IJP ODFs and 100.2–116.3% for OPSs when data from all batches are included.

EXT was found to be the method that fulfilled the criteria for AV for most of the doses and batches, indicating that the method showed best accuracy and precision regarding uniformity of the active substance among dosage units. The smaller the target dose, the more often the prepared dosage forms prepared by different techniques failed to comply with the AV as displayed in Table A1. Both utilized printing methods were, however, shown to be promising manufacturing methods for production of personalized doses when compared to the method currently used at the hospital pharmacy for preparation of tailored doses.

EXT ODFs revealed a linear correlation between the dry weight of the film and the drug content (mg) in the film. Batch 1, 2, and 3 were found to have R^2^ values of 0.9996, 1, and 0.9999, respectively, suggesting that the weight of the dry EXT ODF could be used as an easily accessible tool for quality control in a hospital setting. EXT showed AV values ≥ 40 for two out of the three printed batches of the smallest ODF (target dose of 0.1 mg). This could, however, be anticipated already when calculating the drug concentration in the printing solution as the calculations revealed that the concentration should have been lower for the smallest size than for the rest of the sizes (Table 3). This suggests that to achieve acceptable AV values for the smallest sized EXT ODFs, an optimization of the g-code, design, or the drug concentration of the printing solution should preferably be made, while the other sizes may be printed with the used settings and drug concentration. It was additionally observed that the wet weight of the printed ODFs would deviate from the normal at the beginning of the printing session. This could also be an explanation to why the smallest size EXT ODFs showed high AV, as this strength was the first to be printed in each batch. However, additional studies would be needed to investigate this in a more structured manner.

During IJP of batch 1, some problems were observed as the ink was not stable in the printer before the printing process was started, which could be seen as disappearing droplets during the printing step. This may be explained by (partial) drying of the print head between printing session as it has been seen that the used print head works best if it constantly is kept wet. The printing process proceeded smoothly for batch 3, where the same nozzles could be used for printing of all different sizes and sheets, which directly shows in the drug content results as generally small AV. IJP has previously been described as a method suitable for manufacturing of low-dose dosage forms. However, in many conducted studies, only a single or a few nozzles have been used [53,54,55] compared to 40–60 nozzles used in this study, which will have an impact on the manufacturing time and further may impact the drug content. Independent of the amount of nozzles used, dosage forms with a drug amount deviating from the theoretical calculated amount have been reported [56]. As the dpi, used to obtain the target dose, is calculated based on the droplet size of the jetted droplets, a change in the average droplet size will directly have an impact on the amount of drug deposited in a certain area. In this study, the approach to utilize a high number of nozzles was preferred to decrease the printing time. However, as modest optimization regarding the ink and printing parameters was conducted in this study, the printing results were seen to differ depending on the nozzles used (placement in the print-head), especially during printing of batch 1. Furthermore, with an increased number of nozzles utilized a greater possibility of variation in droplet volume is introduced. Therefore, close attention should be paid to the droplet size of the used nozzles. Further development and optimization of the printing parameters to obtain droplets with a standard deviation as small as possible would likely improve the results for the IJP dosage forms. Moreover, the droplet volume of all nozzles intended to be used should preferably be investigated in a more systematic and automated approach than what was done in this study, where nozzles deviating too much from the average droplet volume would be excluded.

Regarding the optimization of the OPS, pure drug together with the filler lactose monohydrate could be used instead of the Marevan tablet. This would enable a more homogeneous blend as the particle size of the two materials would be more similar. Alternatively, improved grinding of the tablets and blending of the final mixture in a standardized manner would likely help to improve the content uniformity as well as decrease batch to batch variability.

The study revealed that even though the accuracy in some cases was a bit off, the precision was still excellent, seen as a small standard deviation of the drug content within the same batch. This indicates that the printing techniques are suitable techniques for manufacturing of personalized dosage forms even though some optimizations regarding dosage form design, printing parameters or formulation may need to be carried out to further excel the results. As a conclusion, the innovative ODFs prepared by means of printing were shown to be equally good and even improved with regard to uniformity of dosage units, than the OPSs currently used in a hospital setting.

The stability study revealed that the drug amount was kept unchanged during the studied period, even though small fluctuations were seen between the weeks that most likely originates from normal standard deviations within one batch (Figure 7). All the dosage forms with a target dose of 2 mg fulfilled the UC throughout the one-month period (Table A5). Furthermore, both the EXT ODFs and IJP ODFs showed satisfactory acceptance values, whereas the OPSs displayed slightly too high AV values for each week. This suggests that all of the prepared dosage forms may be used after storage at least for one month in a non-controlled environment, as typically is the case in a hospital ward.

### 3.7. In Vitro Dissolution

To date, neither the Ph. Eur. or United States Pharmacopeia (USP) specify the dissolution setup nor the requirements for fulfillment of the developed ODF, making it difficult to compare results from different studies. This is a known dilemma in the field of novel ODFs, and to further emphasize this, Speer et al. have recently demonstrated how different conditions and dissolution setups affect the in vitro drug release properties of ODFs [57].

In this study, the manually conducted in vitro dissolution studies revealed that all dosage forms with a target dose of 2 mg released 80% of the drug within the first 30 min (Figure 8). Both EXT and IJP ODFs displayed a similar drug release behavior. As expected, the OPS was the formulation with the fastest drug release, as the drug in this formulation was present in ground tablets, thus enabling a rapid release of the water-soluble drug due to a large surface area. On the contrary, for the ODFs, the drug present on the surface of the dosage form could immediately be released, while the rest was released once the polymer network was ruptured. When HPC particles undergo hydration, a viscous gel layer is formed, which inhibits further wetting from the inside and resulting in a slower drug release [47]. The thin manufactured ODFs were seen to disintegrate quickly. However, a complete drug release was not observed until around 30 min, leading to the suggestion that the drug particles after the ODF ruptured into smaller pieces, still were embedded in the polymer matrix. The size of the ODFs or OPSs were not observed to have a pronounced impact on the drug release, which is in line with the results obtained in the disintegration study.

When the on-line dissolution was conducted in 500 mL of purified water, the dissolution of the drug was faster for all formulations due to the increased amount of liquid and improved stirring compared to the manual setup. The EXT ODF and OPS with a target dose of 2 mg displayed an 80% drug release within the two first min of the experiment. Corresponding drug release for the IJP ODF was 4 min (Figure 9). This underpins the difference in results gained from the different setups as previously discussed. An automated setup is a more robust method due to decreased human errors and would, therefore, be favorable. However, the on-line dissolution setup could not be used for the smallest dosage forms in this study due to a low dose in the dosage form and a large volume of media required. A harmonized dissolution method tailored for ODFs would, therefore, be desired to excel the research of ODFs.

### 3.8. Evaluation of Drug Administration through a Naso-Gastric Tube

ODFs are normally administered directly into the mouth where they rapidly disintegrate. However, in hospital wards, there might occasionally be a need for the dosage form to be administered through an enteral feeding tube. Therefore, it must be possible to dissolve the dosage form in a small amount of water and subsequently administer it through the naso-gastric tube without the dissolved formulation blocking the tube. The simulation of drug content passing through the naso-gastric tube was performed to investigate the drug amount obtained after administration through the tube. The average amount of WS passing through the tube was compared to the average content measured for the same batch of each dosage form. The average drug amount passing through the naso-gastric tube was 92% for the OPSs, 84% for EXT ODFs and 75% for IJP ODFs (Table 6). Previous studies have shown, that from OPSs containing the drug dipyridamole and lactose as a filler, the amount of drug passing through a naso-gastric tube was 77.5–86.1% depending on the particle size of the filler (< 355 µm vs. < 250 µm) [35]. In size 0 gelatin capsules containing WS and lactose (particle size < 355 µm), the corresponding amount was 96.4%. The use of celluloses (MCC and SMCC) as filler increased the drug loss and occasional blockage of the tube was identified. In this study, no visual blockage of the tube was observed. However, the slightly lower amounts of WS passing through the naso-gastric tube for printed dosage forms suggest that the viscosity of the dissolved HPC films made the solution stick to the surface of the cup and tubing to a higher extent than for the lactose-containing OPSs. The prepared IJP ODFs contained an increased amount of polymer due to increased thickness compared to the EXT ODFs. This in combination with the fact that the IJP ODFs additionally contained PG, seem to have an impact on the drug amount passing through the naso-gastric tube in the studied setup. Some of the drug loss can also be explained by the fact that the syringe tip did not reach every corner of the cup, suggesting that interpersonal differences will occur when administering the dose in this manner.

### 3.9. ATR-FTIR

The ATR-IR spectra of pure substances, physical mixtures as well as OPS, drug-loaded and placebo EXT and IJP ODFs are presented in Figure 10. The physical mixture of HPC and the drug showed bands attributed to HPC (3431 cm^–1^, 2969 cm^–1^) [58] as well as WS (1663 cm^–1^, 1453 cm^–1^, 1323 cm^–1^, 1720 cm^–1^, 759 cm^–1^ and 704 cm^–1^) [32,59,60]. For the drug-loaded EXT ODF, similar bands as to pure WS were seen at 1326 cm^–1^, 701 cm^–1^, 760 cm^–1^, and the broad peak characteristic for pure HPC was present at 3600–3100 cm^–1^ [61]. The placebo EXT ODF revealed the broad peak ranging from 3600–3100 cm^–1^ and further bands at 1452 cm^–1^ and 1326 cm^–1^, suggesting that these may not be attributed to WS in the drug-loaded EXT ODF. As expected, the spectra of the placebo IJP ODF and placebo EXT ODF were identical as the formulation was the same in both placebo dosage forms, only the manufacturing process into ODFs differed. Moreover, drug-free IJP ODFs imprinted with placebo ink showed identical spectra as the placebo IJP ODF prior to printing, even though new components, such as PG and colorant were present in the imprinted placebo sample, leading to the suggestion that the difference in spectra between the drug-loaded IJP ODF compared to the drug-loaded EXT ODF may be a result of some interaction between the drug and the components in the ink. Bands attributed to WS were seen at 759 cm^–1^ and 700 cm^–1^. The only characteristic band for WS identified for the OPS was located at 760 cm^–1^. This band was, however, also found for pure lactose monohydrate. The spectra for the OPSs as well as the Marevan tablet were almost identical to the spectra of lactose monohydrate, which may be explained by the fact that lactose monohydrate was in all formulations present in a much larger ratio than the drug.

No significant spectral shifts, differences in band intensity, or completely dilution of bands were observed for any of the dosage forms during the stability study, indicating that no major change in intermolecular interaction occurs during storage up to one month.

### 3.10. DSC

The thermal properties of the raw materials, physical mixtures, and final dosage forms were investigated once for the raw materials and over a one month period for the prepared dosage forms in order to gain information about how the thermal properties of the dosage forms change upon storage. WS showed a small and broad endothermic event with an onset at 36.5 °C and peak max at 74.1 °C followed by another larger endothermic event identified as the melting of the drug with onset and peak max at 177.7 and 192.7 °C, respectively (Figure 11).

The placebo EXT ODFs revealed a broad peak in the range of around 20–100 °C attributed to dehydration of water from the hygroscopic polymer followed by a small melting peak with an onset at 170.6 °C and a peak max located at 190.6 °C, which correlates to pure HPC prior to processing. Drug-loaded EXT ODFs revealed similar endothermic event, revealing a melting onset at 168.1 °C and a peak max at 185.0 °C. The addition of WS in the formulation resulted in a slight melt point depression compared to pure polymer, which may be explained by the interaction of the drug and the polymer leading to a reduction of the chemical potential of the system. However, no evident melting peak for the drug was seen, suggesting that the drug was not present in a crystalline form. The thermographs for IJP placebo ODFs were as expected, similar to the placebo EXT ODF and pure HPC, indicating that the solvent casting process does not affect the thermal properties of the polymer. Imprinting placebo IJP ODFs with drug-loaded ink resulted in broadening of the second endothermic event, which typically started immediately after the first one. The melt point depression and broadening of the melting event may be explained by the additional materials present in the IJP ODFs compared to the EXT ODFs such as PG acting as a plasticizer in the formulation. The OPSs revealed a sharp melting peak onset at 141.6 °C and peak max at 147.3 °C followed by decomposition of the material. The endothermic events correspond to lactose monohydrate used as a filler in the OPSs. Lactose monohydrate has been reported to lose the crystallization water at temperatures above 100 °C, complemented by a change in the crystalline structure of the material as it becomes anhydrous [50]. At 140 °C, the loss of water is completed, whereafter the material decomposes, thus explaining what was seen on the thermograms for the OPS. Due to the presence of lactose monohydrate in the formulation for the OPSs, no further conclusions could be drawn regarding the drug as the formulation decomposed prior to the melting of WS.

The thermograms of the different dosage forms prepared were not seen to change during storage for four weeks leading to the conclusion that the dosage forms may be given an expiration date of at least one month after manufacturing, which also is supported by the results in the drug content section.

### 3.11. Stability

A stability study was conducted over four weeks in order to assess the stability of the prepared dosage forms. OPSs were stored in sachets, which is how they are stored at the hospital, whereas single ODFs were not packed in a final packaging, exposing them to fluctuating humidities and temperatures. These conditions were thought to mimic the worst case scenario of how extemporaneously compounded on-demand medicines could be stored at the hospital wards. As the idea of personalized medicines is that small batches are prepared due to possible changes in the treatment, one month was considered to be a long enough follow-up period. All of the prepared dosage forms were found to be stable or possess acceptable properties during the one-month long stability study. The dosage forms were subjected to temperatures and relative humidities ranging from 19.6 to 21.5 °C and 13.0 to 32.9%, respectively. More detailed information regarding the results from the stability study is discussed in the different results sections of the manuscript.

### 3.12. Suitability of Manufacturing Methods in a Hospital Pharmacy Setting

The evaluation of the suitability of two different printing technologies as manufacturing methods for extemporaneously prepared medicines in hospital pharmacy setting was made based on aspects of patient safety, manufacturability, ease of administration, and GMP compliance adapting perspectives found in the literature [22,23]. Both advantages and limitations were identified with all manufacturing methods and dosage forms prepared in this study.

Recognized advantages for all investigated dosage forms in this study were the results of content uniformity and the one-month stability. EXT further showed promising results for uniformity of dosage units, and both EXT and IJP ODFs displayed good mechanical properties and fast disintegration. All dosage forms were suitable for administration through a naso-gastric tube, whereas the ODFs additionally have the option to be administered directly into a child’s mouth. The inkjet printing method was successfully used to imprint QR-codes onto the previously prepared EXT and IJP ODFs, thus, likely improving patient safety by enhancing the identification of the dosage form.

Requirements for utilizing printing techniques in hospital pharmacy environment is the use of pharmaceutical grade excipients and printers that fulfill the demands of good manufacturing practice (GMP) [24]. One such aspect is the cleaning of the printer parts that come into contact with the pharmaceutical product. In EXT it is possible to use disposable parts thus avoiding cleaning procedures and validations. The IJP printer used in this study needs to be flushed with a suitable solvent after use in order to ensure that contamination between different drugs and/or formulations do not occur. Separate print heads (IJP) for specific drug solutions might decrease the burden of cleaning in cases where the same printer is used for different formulations or drugs. Moreover, disposable ink cartridges and tubing or alternatively stainless steel parts that may be cleaned would be compulsory in order to be able to transfer the printers from laboratories into hospitals. As for all medicines, the safety of excipients for use in neonates and infants must be taken into consideration as well as the suitability of the formulation for administration through a naso-gastric tube without causing blockage of the tube. The solvents ethanol and PG used in this study are excipients that might cause adverse reactions in children, especially neonates and infants, and the use of such excipients should be based on a risk assessment [62]. In future studies, PG could be replaced with another viscosity modifier and residual amounts of ethanol in the dosage forms could be measured to better evaluate the risk as in this study, the residual solvent was not analytically determined, but instead theoretically calculated to be well below the permitted daily exposure of 50 mg/day for ethanol [63]. Excipient availability from reliable suppliers is also important from GMP and safety point of view as well as for assuring the continuous supply of extemporaneously prepared dosage forms. The transparency sheet, as part of manufacturing equipment, could be replaced by a printing platform made of, e.g., glass, which could be easily removed and cleaned to fulfill GMP requirements.

Investment costs and annual costs for maintenance and qualification are factors that might have an impact when deciding on the choice of printing method to be implemented in a hospital. For on-demand manufacturing purposes, it would be necessary to have duplicates of devices to ensure a continuous supply of printed dosage forms even in case of malfunction. Devices for high-end IJP equipped with a camera needed for analysis of droplets are more expensive than the basic devices used for EXT. The training necessary for pharmacists using the devices and the batch specific premanufacturing tests ensuring the correct function of the devices is probably easier to perform for the EXT than the IJP, at least with the printers utilized in this study. In EXT, one could print test dosages and weigh them prior to printing of patient doses as an excellent correlation between the wet weight and the drug content was shown in this study. For IJP one has to test the functioning of all nozzles before printing, which may be time-consuming, but can be automated. Hence, in the future devices for IJP should have an automatic quality check of the nozzles, excluding nozzles deviating from the set droplet volume range. It is worth noticing that laborsome and time-consuming quality checks (especially manually performed) before manufacturing adds to the total manufacturing time.

Both printing techniques involve some manufacturing steps concerning the formulation that needs to be solved in order for the methods to be suitable for on-demand manufacturing. The printing solution used in both EXT and IJP should be ready-made and kept in stock as the manufacturing of these solutions required stirring overnight. Another option would be to keep the polymer solution in stock and add the API immediately before printing, which could be beneficial as the same drug-free print solution could be used as a base formulation for different APIs. As for the IJP method, the substrates should also be kept in stock enabeling on-demand manufacturing. Another time-consuming step in the overall printing process is the drying of the printed drug product, which can be optimized, e.g., by using suitable technical heating and drying solutions. The drying phase should ideally be decreased to much less than an hour in order to make printing a suitable method for manufacturing personalized medicines in hospital pharmacy settings where the time from prescribing to delivery is kept as short as possible. The production of a batch of OPSs takes one hour, at most. Further, studies should be conducted to evaluate suitable drying methods for printed ODFs to decrease the total manufacturing time.

Limitations that are essential for adopting a new manufacturing technique into the hospital pharmacy setting should be resolved before the method transfers from a laboratory environment into hospital pharmacies. In this study, we have discussed the advantages of the printing of dosage forms and showed that the ODFs prepared by means of printing in many aspects were superior to the OPSs currently used at hospitals. However, the present study also underlines the critical aspects that still need to be resolved in order to accept novel printing techniques as manufacturing methods in hospitals. Based on this study, we believe that method and product development, as well as further optimization, should be done in close collaboration between academic research labs, hospitals, and regulatory authorities in order to further excel the innovative printing methods to fit the demands of a hospital pharmacy.

## 4. Conclusions

Tailored drug doses are of great importance for successful and safe therapies, especially for pediatric patients. This study compared an established manufacturing technique to manufacture OPSs with two novel printing techniques (EXT and IJP) for preparation of ODFs. Dosage forms of various strengths of WS were successfully produced utilizing all of the studied manufacturing methods. The prepared ODFs showed acceptable properties and were found to be superior to the established OPSs regarding uniformity of dosage units, proving that the investigated printing techniques are precise methods that in the future could be implemented in a hospital setting for the preparation of personalized doses. The ODFs prepared by printing may, furthermore, have an advantage in ease of administration for pediatrics and children as they can be administered directly in the mouth of the patient, without the need of water, whereas the OPSs would need to be dissolved or dispersed in a liquid prior to administration. In a hospital setting, both dosage forms can alternatively be administered through a naso-gastric tube in case the patient previously has one in place for, e.g., nutritional purposes or in situations where the patient is unconscious. The stability study revealed no loss in quality for the prepared dosage forms during the studied period, independent of the manufacturing technique used.

The present study successfully implemented QR codes directly on the printed ODFs allowing for numerous possibilities such as additional information and increased patient safety. This study, among other recent studies in the field, have shown the feasibility and potential of using printing techniques for manufacturing of flexible doses, contributing to safer and improved treatments for various patient groups in the future. In order to produce personalized on-demand dosage forms for children in a hospital pharmacy setting, special attention should be paid to the safety of used excipients, implementation of suitable non-destructive and fast quality assurance methods. Furthermore, the possibility to use disposable parts instead of time-consuming cleaning procedures and short turnaround time for the complete manufacturing process including printing solution preparation and drying time of final dosage form should be ensured in order to successfully implement printing methods as a part of the manufacturing techniques used in a hospital pharmacy.

## Figures and Tables

**Figure 1 pharmaceutics-11-00334-f001:**
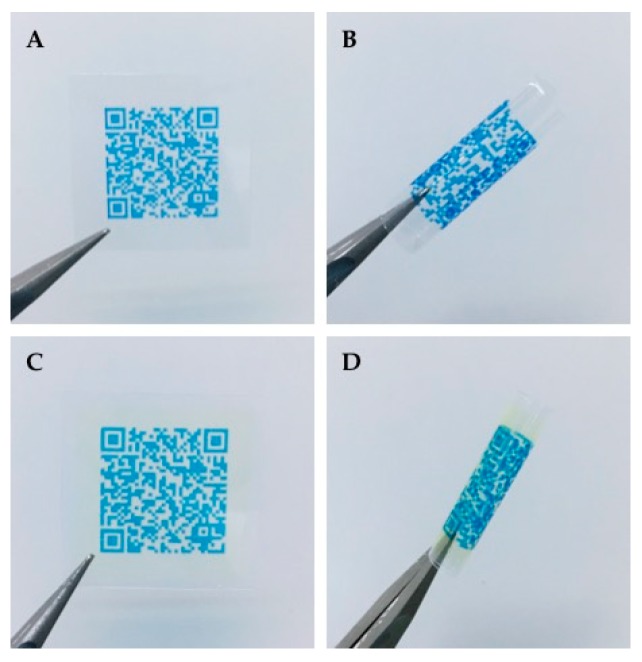
(**A**) EXT drug-loaded ODF imprinted with a QR code containing information about the dosage form and (**B**) the same EXT ODF rolled up to visualize the flexibility of the film. (**C**) IJP drug-loaded ODF with a printed QR code and (**D**) the flexible ODF is subsequently coiled up for illustrative purposes.

**Figure 2 pharmaceutics-11-00334-f002:**
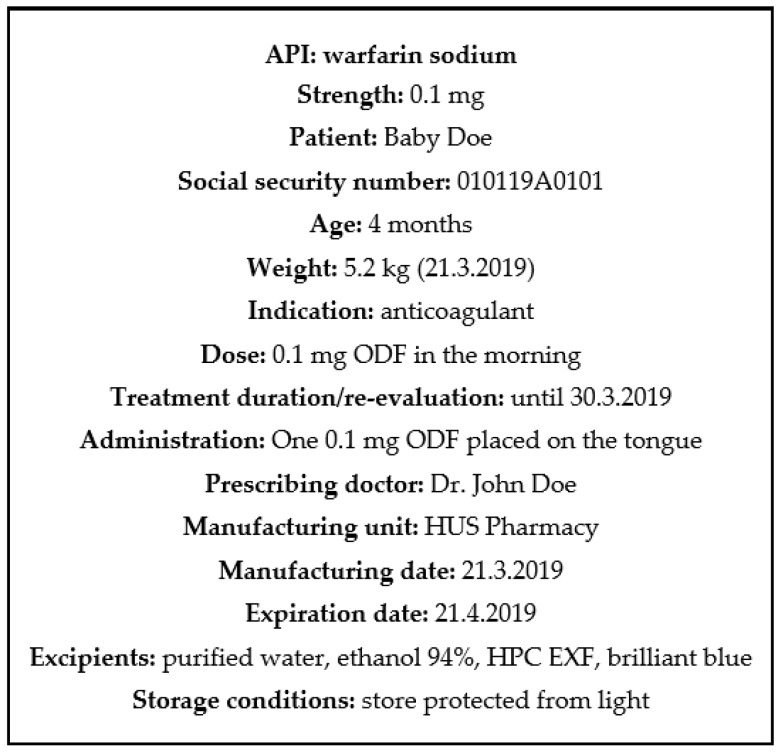
Example of the information that could be included in the QR code.

**Figure 3 pharmaceutics-11-00334-f003:**
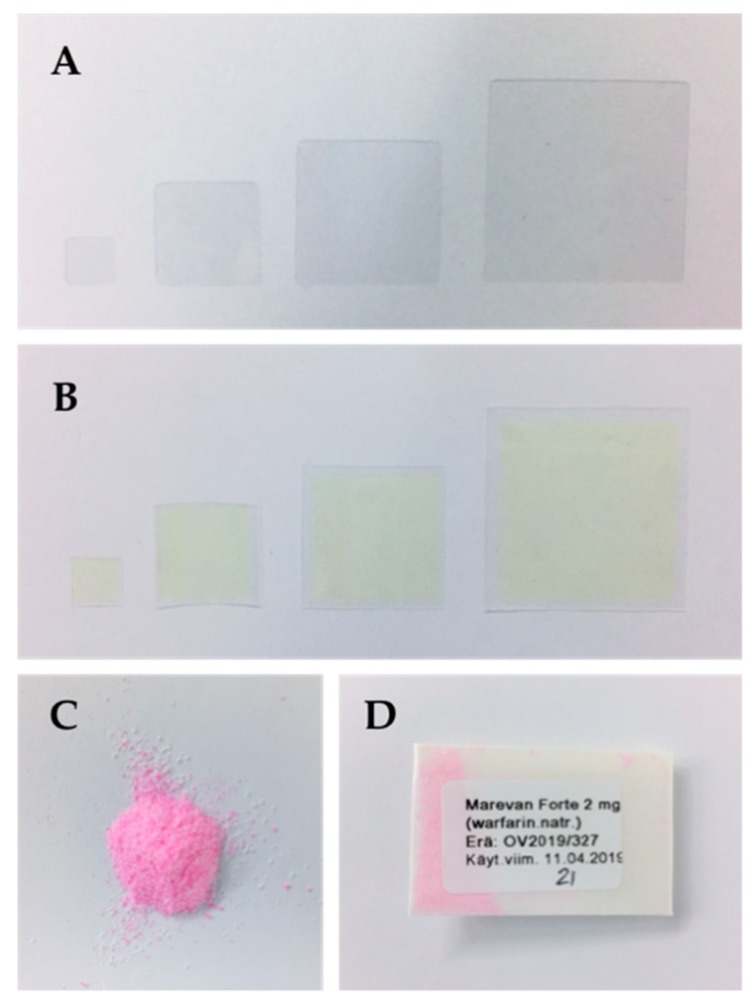
Pictures of the prepared dosage forms: (**A**) EXT ODFs; (**B**) IJP ODFs; (**C**) oral powder; and (**D**) OPS.

**Figure 4 pharmaceutics-11-00334-f004:**
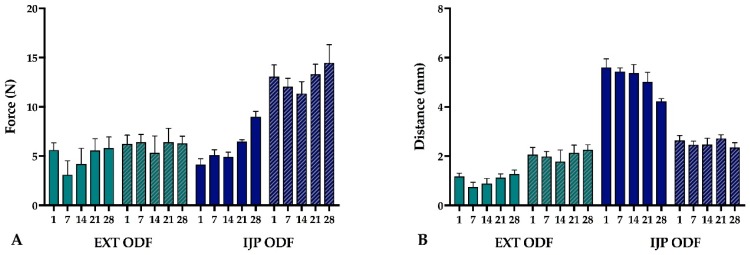
(**A**) Burst strength representing the maximum force (N) and (**B**) burst distance representing the elongation (mm) of the ODF before rupturing. Drug-loaded ODFs are shown as colored columns and placebo ODFs are presented as colored columns with a pattern. Data is shown as average ± SD, *n* = 5 for each time point.

**Figure 5 pharmaceutics-11-00334-f005:**
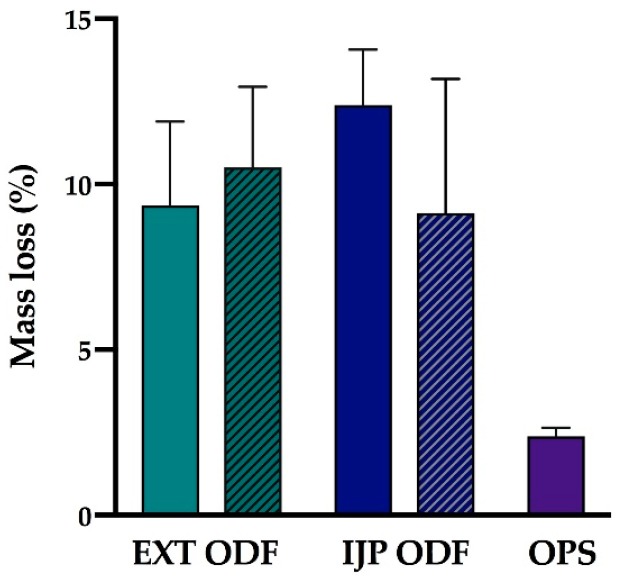
The moisture content reported as mass % weight loss of the sample, for the prepared dosage forms. Results presented as average ± SD, *n* = 3. Drug-loaded ODFs are shown as colored columns and placebo ODFs are presented as colored columns with a pattern.

**Figure 6 pharmaceutics-11-00334-f006:**
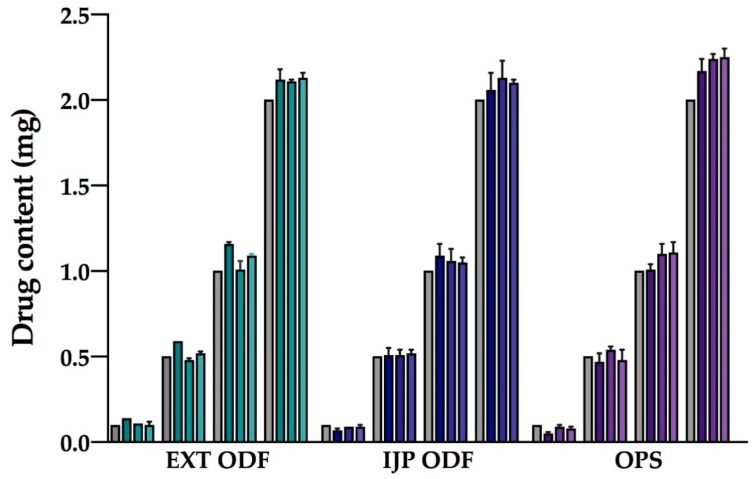
Drug content of the aimed doses of 0.1, 0.5, 1, and 2 mg for the prepared batches and various manufacturing techniques. Gray columns represent the target dose and the following columns represents batch 1, 2, and 3, respectively, for each manufacturing method. Data are presented as the average ± SD, *n* = 10.

**Figure 7 pharmaceutics-11-00334-f007:**
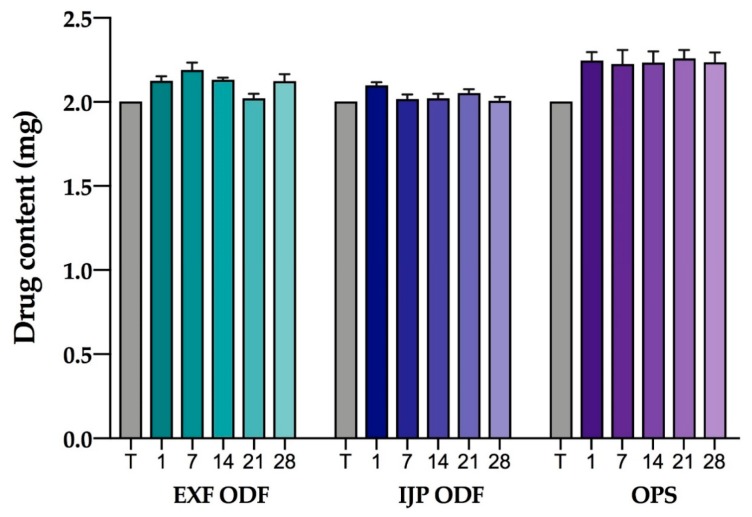
Stability of the manufactured dosage forms with a target dose of 2 mg at time points 1, 7, 14, 21, and 28 days. The gray columns represent the target dose of 2 mg. Data shown as average ± SD, *n* = 10.

**Figure 8 pharmaceutics-11-00334-f008:**
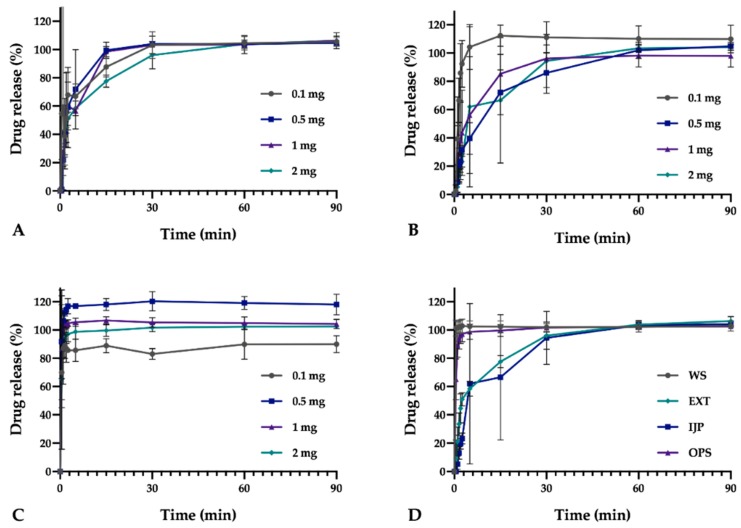
In vitro drug release of the prepared dosage forms in 100 mL of purified water, average ± SD (*n* = 3); (**A**) EXT ODFs, (**B**) IJP ODFs, (**C**) OPSs, and (**D**) an overview of all manufacturing methods and pure drug for the 2 mg dose.

**Figure 9 pharmaceutics-11-00334-f009:**
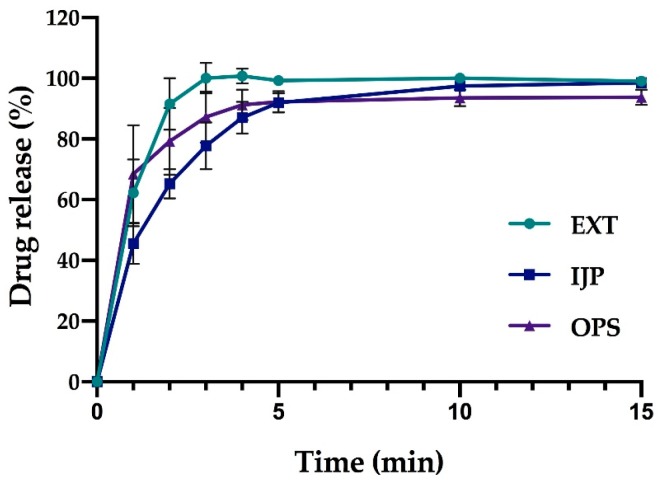
On-line in vitro drug release (%) of the different dosage forms conducted in 500 mL of purified water at 50 rpm, average ± SD, *n* = 3.

**Figure 10 pharmaceutics-11-00334-f010:**
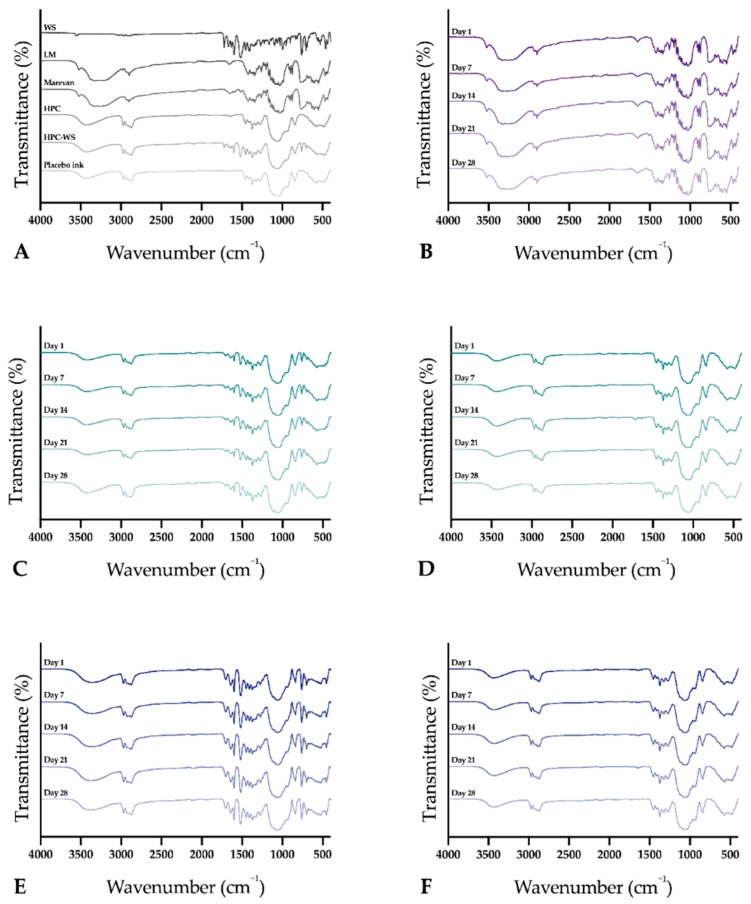
FT-IR spectra for (**A**) raw materials and/or starting materials; (**B**) OPS; (**C**) drug-loaded EXT; (**D**) placebo EXT ODF; (**E**) drug-loaded IJP ODF; and (**F**) placebo IJP ODF. The graphs present the spectra for the prepared drug-loaded and placebo dosage forms for day 1, 7, 14, 21, and 28.

**Figure 11 pharmaceutics-11-00334-f011:**
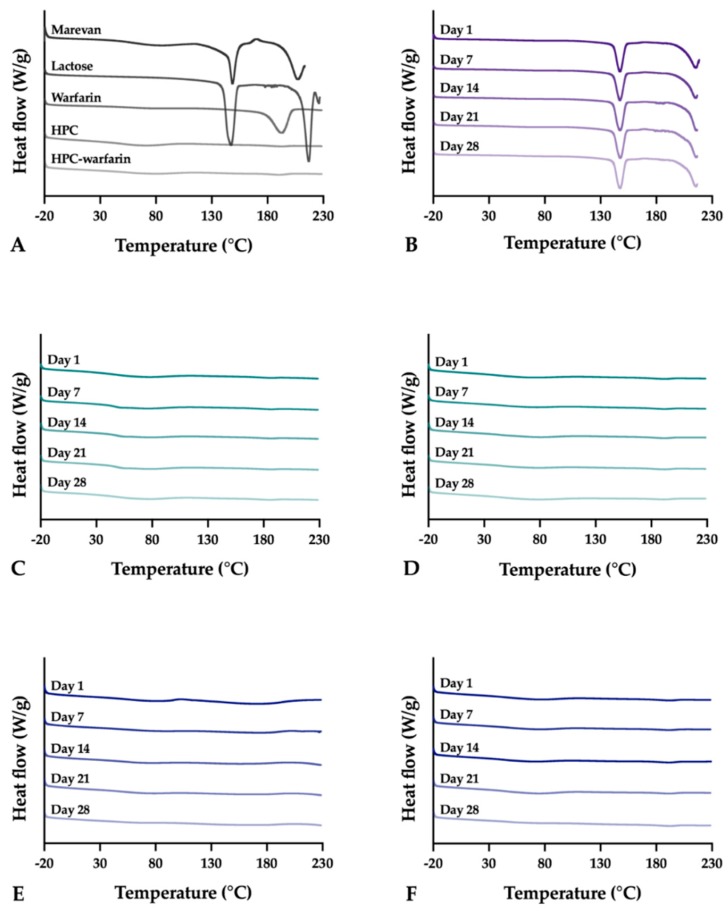
Thermograms (exo up) for (**A**) raw materials and the physical mixture of HPC and warfarin sodium; (**B**) OPS; (**C**) drug-loaded EXT ODF; (**D**) placebo EXT ODF; (**E**) drug-loaded IJP ODF; and (**F**) placebo IJP ODF over a one-month period.

**Table 1 pharmaceutics-11-00334-t001:** Designed geometries for the ODFs.

Target Dose (mg)	Total Film Length (mm)	EXT ODF		IJP ODF
		Height (mm)	Volume (mm^3^)	Printed Length (mm)
0.1	5 × 5	0.1	2.5	4.4 × 4.4
0.5	11.2 × 11.2	0.1	12.5	9.8 × 9.8
1	15.8 × 15.8	0.1	25	13.9 × 13.9
2	22.4 × 22.4	0.1	50	19.7 × 19.7

**Table 2 pharmaceutics-11-00334-t002:** Ink compositions for IJP.

Substance	Drug-Loaded Ink (*w*/*w* %)	Placebo Ink (*w*/*w* %)
Warfarin sodium	10	-
Quinoline yellow	0.01	0.01
PG	27	27
Water	5	5
Ethanol	Ad 100	Ad 100

**Table 3 pharmaceutics-11-00334-t003:** Wet and dry weights of the placebo EXT ODFs used to determine the drug load in the printing solution.

Target Dose	Wet Weight (mg)	Dry Weight (mg)	Theoretical Drug Concentration (%)	
0.1	7.7 ± 0.1	1.2 ± 0.1	1.3	
0.5	32.0 ± 0.2	5.3 ± 0.1	1.6	
1	63.6 ± 0.5	10.5 ± 0.1	1.6	
2	124.9 ± 1.2	21.4 ± 0.1	1.6	
R^2^	1	0.9998	Average	1.5

**Table 4 pharmaceutics-11-00334-t004:** Manufacturing times for EXT ODFs and IJP ODFs. The manufacturing time includes the actual printing time, not premanufacturing steps nor drying times of films. For inkjet printing 51 ± 9 nozzles were used for target doses 0.1, 0.5, and 1 mg and 45 ± 7 nozzles for a target dose of 2 mg.

Target Dose	Manufacturing Time	
	EXT ODF	IJP ODF
0.1	42 s	3 ± 1 s
0.5	1 min 9 s	7 ± 1 s
1	1 min 44 s	13 ± 2 s
2	2 min 53 s	18 ± 3 s

**Table 5 pharmaceutics-11-00334-t005:** Surface pH of the drug-loaded 2 mg and corresponding placebo dosage forms after 1 min and 15 min, respectively, average ± SD, *n* = 3. WS = warfarin sodium, P = placebo.

Sample	Surface pH	
	1 min	15 min
EXT WS ODF	7.13 ± 0.13	7.07 ± 0.05
EXT P ODF	6.94 ± 0.10	6.43 ± 0.28
IJP WS ODF	7.37 ± 0.28	7.05 ± 0.03
IJP P ODF	7.03 ± 0.23	6.35 ± 0.21
OPS WS	9.36 ± 1.62	7.34 ± 0.19

**Table 6 pharmaceutics-11-00334-t006:** Content of warfarin sodium passing through the naso-gastric tube after dissolving/dispersing the different dosage forms in 2 mL of water. Results presented as the average ± SD, *n* = 3.

Dosage Form	Average Content (mg)	Content Compared to Average Batch Content (%)
EXT ODF	1.78 ± 0.05	84
IJP ODF	1.57 ± 0.12	75
OPS	2.07 ± 0.06	92

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
