# Peer review of "Towards Printed Pediatric Medicines in Hospital Pharmacies: Comparison of 2D and 3D-Printed Orodispersible Warfarin Films with Conventional Oral Powders in Unit Dose Sachets"

_pharmaceutics, 2019, doi:10.3390/pharmaceutics11070334_

Reviewer 1 Report

The manuscript discusses the potential of making personalised dosage forms by 2D/3D printing in a hospital setting. The content is inline with actual concerns and can greatly benefit the community as an example of what type of investigation needs to be done to address this concern. The manuscript is well written and easy to read and i recommend its acceptation for publication after some minor modifications.

Some tables ( 4;6;7; 8 and 9) can be put in supplementary data to lighten the manuscript.

L128 replace EXP by EXT

L199 : how was residual solvent assessed

L283-285 : better define end point of disintegration test ( complete dissolution or not and if not what type of film pieces or particles are seen/assessed)

L597-603 : Authors state moisture to be necessary for mechanical properties through it’s plasticizing effect, they should mention other potential plasticizers which would have maybe less stability issues than water.

L743: explain reason of drop instability and possible solutions

Author Response

Dear reviewer, 

We are very thankful for your constructive comments and suggestions for improvement of the manuscript. We have answered/commented all questions/suggestions in a point-by-point manner below and made changes to the text accordingly. We have marked the changes with track-changes in the resubmitted manuscript. 

The manuscript discusses the potential of making personalised dosage forms by 2D/3D printing in a hospital setting. The content is in line with actual concerns and can greatly benefit the community as an example of what type of investigation needs to be done to address this concern. The manuscript is well written and easy to read and i recommend its acceptation for publication after some minor modifications. 

Some tables ( 4;6;7; 8 and 9) can be put in supplementary data to lighten the manuscript. 

Thank you for pointing this out, we have now moved tables 4, 6, 7, 8, and 9 to the supplementary section and made changes in the manuscript accordingly. 

L128 replace EXP by EXT 

This is now corrected, thank you for pointing out the mistake. 

L199 : how was residual solvent assessed 

Good point- residual solvent was unfortunately not analytically assessed in this study, but was calculated, based on the dry weight of the dosage form, to be well below the limit for the recommended daily intake of ethanol.  

L283-285 : better define end point of disintegration test ( complete dissolution or not and if not what type of film pieces or particles are seen/assessed) 

We have now tried to explain the endpoint better in the manuscript. 

L597-603 : Authors state moisture to be necessary for mechanical properties through its plasticizing effect, they should mention other potential plasticizers which would have maybe less stability issues than water. 

Commonly used plasticizers for ODFs are now added to this section. 

L743: explain reason of drop instability and possible solutions 

We are not quite sure why this phenomenon occurred during printing of batch 1 and it should be investigated further. It might be that the ink was not allowed to reach room temperature before starting the printing process (stored in the fridge) or that there was some partial clogging in the print head due to drying of the print head in between printing sessions (the print head works best if it is not allowed to dry). Further studies would, however, be needed to be able to point out the reason to why the ink was unstable during printing of the first batch. A possible explanation is added to the manuscript. 

Reviewer 2 Report

The manuscript entitled “Towards printed pediatric medicines in hospital pharmacies: Comparison of 2D and 3D-printed orodispersible warfarin films with conventional oral powders in unit dose sachets” presents very interesting and comprehensive study on the possible utilization of 3D printing in hospital pharmacy practice. In general, the manuscript is well written, the results are well-presented and the conclusions are very interesting from the practical point of view. However some minor improvements could be made. Therefore, I have some comments listed below:

1.      The article is very long, it seems that some parts can be shortened:

a.      Obtained manufacturing times are discussed excessively, because in the hospital pharmacy where small batches or individual dosage forms are printed it is not so important as in the large scale production. In my opinion this section should be shortened. Results would be more legible in the form of graph or table with just a 3 -4 sentences of comment.

b.      ATR-FTIR and DSC studies didn’t revealed any interesting findings. Authors indicated that many results were expected. Such a detailed description of the events in the DSC or bands in the ATR-FTIR spectra characteristic for substances seems to be unnecessary. Thus, those sections can be shortened to just a few sentences of comments with focus on the most important findings, and it won’t decrease the quality of the manuscript. The authors should mainly focus on findings which are unobvious.

2.      The use of the “ink” term in case of extrusion-based printing (line 166 and further) is quite confusing, especially when there is a second printing method such as inkjet printing in the same work. I would advise to use other term to differentiate the 3D printing gel which is extruded, from ink which is deposited on the surface in the 2D printing process.

3.      Some data are duplicated i.e. the burst strength and distance presented in the figure 4 can be also found in table 4. The same situation is in the figure 6 and table 8 for drug content. Taking into account that the manuscript is excessive I would advise to delete the figure 4 and 6 and to make necessary corrections in the text.

4.      The dissolution methodology and results raise doubts. The standard deviations are huge for some formulations, thus the number of tested samples should be increased to at least 6. The reason of this large SD values may be the method of dissolution chosen by authors. I understand that dissolution of 0.1 mg to 500 mL could be an analytical problem but authors can always use a cuvette with longer optical path or other analytical procedure with higher sensitivity e.g. HPLC. Small volume dissolution chambers, which are available for Sotax AT7 dissolution tester may also be used to improve results reproducibility.

Author Response

Dear reviewer, 

We are very thankful for your constructive comments and suggestions for improvement of the manuscript. We have answered/commented all questions/suggestions in a point-by-point manner below and made changes to the text accordingly. We have marked the changes with track-changes in the resubmitted manuscript. 

The manuscript entitled “Towards printed pediatric medicines in hospital pharmacies: Comparison of 2D and 3D-printed orodispersible warfarin films with conventional oral powders in unit dose sachets” presents very interesting and comprehensive study on the possible utilization of 3D printing in hospital pharmacy practice. In general, the manuscript is well written, the results are well-presented and the conclusions are very interesting from the practical point of view. However some minor improvements could be made. Therefore, I have some comments listed below: 

1.      The article is very long, it seems that some parts can be shortened: 

Obtained manufacturing times are discussed excessively, because in the hospital pharmacy where small batches or individual dosage forms are printed it is not so important as in the large scale production. In my opinion this section should be shortened. Results would be more legible in the form of graph or table with just a 3 -4 sentences of comment. 

Thank you for your comment. We have now slightly shortened the discussion regarding the manufacturing times and inserted the results in a table to make the text easier to read as well as deleted some results. We do, however, feel that the manufacturing times and the discussion around it is important for people interested in implementing a printing method into a hospital pharmacy setting as certain medicines needs to be delivered within a short timeframe. 

ATR-FTIR and DSC studies didn’t revealed any interesting findings. Authors indicated that many results were expected. Such a detailed description of the events in the DSC or bands in the ATR-FTIR spectra characteristic for substances seems to be unnecessary. Thus, those sections can be shortened to just a few sentences of comments with focus on the most important findings, and it won’t decrease the quality of the manuscript. The authors should mainly focus on findings which are unobvious. 

We have now shortened the abovementioned sections. A few sentences seem a bit little and, therefore, we thought about the option to completely remove these results. However, we do use the results to draw conclusions in the manuscript and would, therefore, hope to keep it in this shortened version. 

2.      The use of the “ink” term in case of extrusion-based printing (line 166 and further) is quite confusing, especially when there is a second printing method such as inkjet printing in the same work. I would advise to use other term to differentiate the 3D printing gel which is extruded, from ink which is deposited on the surface in the 2D printing process. 

Thank you for pointing out that this may be confusing to the reader. We have now clarified this in the text and only used the term “ink” for the inkjet printing process. 

3.      Some data are duplicated i.e. the burst strength and distance presented in the figure 4 can be also found in table 4. The same situation is in the figure 6 and table 8 for drug content. Taking into account that the manuscript is excessive I would advise to delete the figure 4 and 6 and to make necessary corrections in the text. 

Thank you for the valuable comment. Based on the comments from both of the reviewers tables 4, 6, 7, 8, and 9 have now been moved to the supplementary section to make the manuscript lighter for the reader. In this way, readers can still find all the detailed data if interested. 

4.      The dissolution methodology and results raise doubts. The standard deviations are huge for some formulations, thus the number of tested samples should be increased to at least 6. The reason of this large SD values may be the method of dissolution chosen by authors. I understand that dissolution of 0.1 mg to 500 mL could be an analytical problem but authors can always use a cuvette with longer optical path or other analytical procedure with higher sensitivity e.g. HPLC. Small volume dissolution chambers, which are available for Sotax AT7 dissolution tester may also be used to improve results reproducibility. 

The standard deviation is indeed quite large for the manual setup which is a common problem for manual sampling. Another reason for the variations might be that the film disintegrates into smaller film pieces and these can be sampled increasing the standard deviation. However, the drug release was not the main focus of this study, where the printing methods were evaluated, and the drug content was seen to be more important. We do agree that a more sensitive method such as HPLC should be used in the future so that the automated setup can be utilized.